# WHEN AND WHY MOMENTUM ACCELERATES SGD: AN EMPIRICAL STUDY

## ABSTRACT

Momentum has become a crucial component in deep learning optimizers, necessitating a comprehensive understanding of when and why it accelerates stochastic gradient descent (SGD). To address the question of "when", we establish a meaningful comparison framework that examines the performance of SGD with Momentum (SGDM) under the *effective learning rates* $\eta_{\mathrm{ef}}$, and offers a holistic view of the momentum acceleration effect. In the comparison of SGDM and SGD with the same effective learning rate and the same batch size, we observe a consistent pattern: when $\eta_{\mathrm{ef}}$ is small, SGDM and SGD experience almost the same empirical training losses; when $\eta_{\mathrm{ef}}$ surpasses a certain threshold, SGDM begins to perform better than SGD. Furthermore, we observe that the advantage of SGDM over SGD becomes more pronounced with a larger batch size. For the question of "why", we find that the momentum acceleration is closely related to *edge of stability* (EoS), a recently discovered phenomenon describing that the sharpness (largest eigenvalue of the Hessian) of the training trajectory often oscillates around the stability threshold. Specifically, the misalignment between SGD and SGDM happens at the same moment that SGD enters the EoS regime and converges slower. Momentum improves the performance of SGDM by preventing or deferring the occurrence of EoS. Together, this study unveils the interplay between momentum, learning rates, and batch sizes, thus improving our understanding of momentum acceleration.

## 1 INTRODUCTION

One key challenge in deep learning is to effectively minimize the empirical risk $f(\boldsymbol{w}) = \frac{1}{N}\sum_{i=1}^{N} \ell(\boldsymbol{w}, z_i)$, where $\ell$ the loss function, $\{z_i\}_{i=1}^{N}$ is the dataset, and $\boldsymbol{w}$ is the parameter of deep neural networks. To tackle this challenge, countless optimization tricks have been proposed to accelerate the minimization, including momentum (Polyak, 1964), adaptive learning rate (Kingma & Ba, 2014), warm-up (Goyal et al., 2017), etc. Among these techniques, momentum, which accumulates gradients along the training trajectory to calculate the update direction, is undoubtedly one of the most popular tricks. Momentum has been widely adopted by state-of-art optimizers including Adam (Kingma & Ba, 2014), AMSGrad (Reddi et al., 2019), and Lion (Chen et al., 2023).

The widespread use of momentum necessitates the understanding of **when** and **why** momentum works, which can either facilitate a good application of momentum in practice, or help building the next generation of optimizers. However, it is more than surprising that neither of when and why momentum works in deep learning is clear, even for the simplest momentum-based optimizer stochastic gradient descent with momentum (SGDM). Specifically, (a detailed discussion is included in Section 2)

- As for **when**, recent studies (Polyak, 1964; Yuan et al., 2016; Defazio, 2020; Cutkosky & Mehta, 2020; Leclerc & Madry, 2020; Smith et al., 2020) have primarily focused on the acceleration behavior of momentum within a limited range of hyperparameters, leaving a holistic view unexplored;

- As for **why**, existing explanations (Smith et al., 2020; Defazio, 2020; Cutkosky & Mehta, 2020) can only account for a portion of experimental observations.

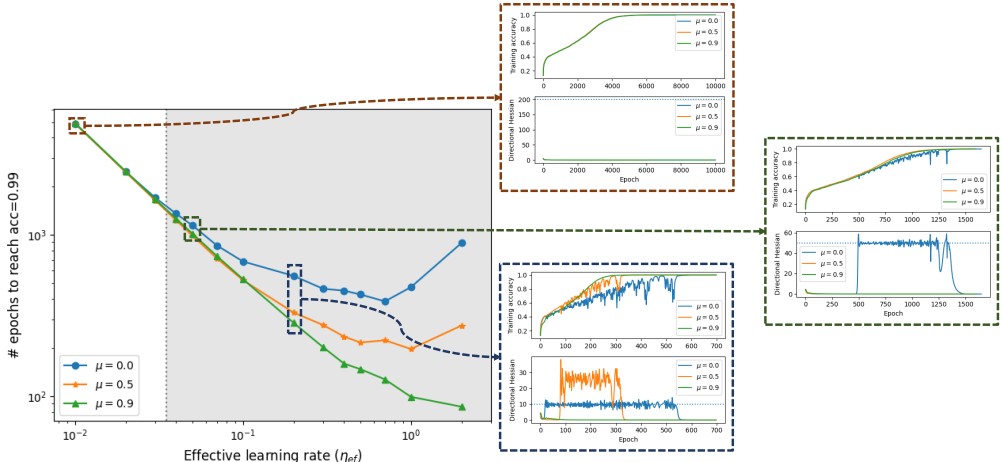

Figure 1: **The slow-down of SGD(M) occurs at the same moment when it experiences abrupt sharpening, i.e., a sudden jump of the directional Hessian along the update direction.** 1) Before experiencing any abrupt sharpening, the training speeds of all optimizers are aligned. The directional Hessian along the update direction for SGD(M) remains small throughout the training process. 2) The convergence of SGD(M) slows down after abrupt sharpening. 3) Momentum defers the abrupt sharpening, thereby helping to accelerate.

In this paper, we investigate the underlying acceleration mechanism of momentum by systematically comparing the performance of SGD and SGDM. The psedo-code of SGD and SGDM [1] is given in Algorithm 1, where SGD is obtained by ignoring the green-highlighted content.

To proceed, we first establish a meaningful comparison framework to avoid unnecessary complexities. Specifically, we compare SGD and SGDM or two SGDMs with different momentum coefficients under the same *effective learning rate*. The effective learning rate is the learning rate multiplied by a factor $1/(1-\mu)$ for a momentum coefficient $\mu$, which takes into account the first-order approximation of the update magnitude induced by momentum. Moreover, we evaluate their performances for a wide range of effective learning rates which cover typical choices.

Our comparison framework gives us a full picture of their performances, which clearly reveals **when** momentum helps acceleration. Based on this framework, we observe a consistent pattern: the training speeds of SGD and SGDM are almost the same for small effective learning rates, but when the effective learning rate increases beyond a certain threshold, SGDM begins to perform better than SGD and shows acceleration benefits. When varying batch sizes, we find that increasing batch size changes the threshold effective learning rate (deviation point), beyond which SGDM performs better than SGD.

To understand why momentum accelerates training, we first focus on the full batch case and study GD(M) since the acceleration effect of momentum is more evident with a large batch size. We find that during the training process, the loss of GD deviates from that of GDM at the same time when GD starts to oscillate. We further attribute the oscillation to a phenomenon of "abrupt sharpening" that the directional Hessian along the update direction first stays around $0$ and then experiences a sudden jump which leads to oscillation. We show abrupt sharpening is a new feature of the renowned concept Edge of Stability, and more importantly, it can be used to theoretically explain the alignment and deviation between GD and GDM: before GD and GDM exhibit abrupt sharpening, the gradient barely change and the updates of GD and GDM are close; abrupt sharpening slows down the convergence; and momentum can defer abrupt sharpening and thus accelerate. We demonstrate this methodology through Figure 1. We further find that a smaller batch size can also defer abrupt sharpening, which overlaps with the effect of momentum. This can explain why decreasing batch size can defer the deviation point.

---

[1] We focus on Polyak's momentum in the main text, and verify that our conclusions also hold for Nesterov's momentum in the Appendix B.

In summary, we empirically investigate the benefit of momentum and our contributions are as follows.

- We introduce a meaningful framework to compare the performances of SGD and SGDM with effective learning rates, which gives a full picture of the momentum benefits.

- **When?** Momentum accelerates SGD when the effective learning rate is larger than a certain threshold, and the threshold will decrease when increase the batch size.

- **Why?** We show that once the optimizer experiences abrupt sharpening, the training process slows down and the momentum can significantly postpone the point of abrupt sharpening.

---

**Algorithm 1** SGD and SGDM

---

1: **Input:** the loss function $\ell(w, z)$, the initial point $\boldsymbol{w}_1 \in \mathbb{R}^d$, the batch size $b$, learning rates $\{\eta_t\}_{t=1}^T$, $\boldsymbol{m}_0 = 0$, and momentum hyperparameters $\{\mu_t\}_{t=1}^T$.
2: **For** $t = 1 \to T$:
3:     Sample a mini-batch of data $B_t$ with size $b$
4:     Calculate stochastic gradient $\nabla f_{B_t}(w_t) = \frac{1}{b} \sum_{z \in B_t} \ell(w_t, z)$
5:     Update $\boldsymbol{m}_t \leftarrow \mu_t \boldsymbol{m}_{t-1} + \nabla f_{B_t}(\boldsymbol{w}_t)$
6:     Update $\boldsymbol{w}_{t+1} \leftarrow \boldsymbol{w}_t - \eta_t \boldsymbol{m}_t$
7: **End For**

---

## 2 RELATED WORKS

Table 1: **Comparison with previous works.** We use the following abbreviations for short words: learning rate → **lr**, batch size → **bs**, difference → **df**, small → **sm**, large → **lg**, accelerate → **acc**, high curvature → **hc** and high noise → **hn**. The noise here refers to the gradient noise. In the table, "**df lr**" indicates exploration done for different learning rate settings, while "**df bs**" refers to different batch size settings. The symbol $(\star)$ represents that momentum accelerates SGD when the effective learning rate is larger than a certain threshold, and the threshold will decrease when increase the batch size. The symbol $(\triangle)$ signifies that momentum accelerates SGD during the early stages of optimization.

| | Non-convex | when | | | Why |
|---|---|---|---|---|---|
| | | Effective lr | Setting | Conclusion | |
| Polyak (1964) | | | **lg bs** (GD) | **acc** | Cancel out effect of **hc** |
| Yuan et al. (2016) | | ✓ | **sm bs** + **sm lr** | don't **acc** | |
| Defazio (2020) | ✓ | | | $(\triangle)$ | cancel out noise |
| Cutkosky & Mehta (2020) | ✓ | | | **acc** under **hn** | cancel out noise |
| Leclerc & Madry (2020) | ✓ | | **df lr** | **acc** under **sm lr** | |
| Smith et al. (2020) | ✓ | ✓ | **df bs** | **acc** under **lg bs** | Cancel out effect of **hc** |
| Wang et al. (2023) | ✓ | ✓ | **sm lr** | don't **acc** | |
| Ours | ✓ | ✓ | **df lr** + **df bs** | $(\star)$ | Prevent from entering EoS |

**When** Our work makes the progress that: **1) Offering a holistic view of the momentum acceleration effect.** Prior studies have either examined the impact of momentum in specific settings or focused on its effects when modifications are made to the learning rate or batch size. In contrast, this work investigates momentum in more comprehensive settings, including the interplay between momentum, learning rate, and batch size. Furthermore, we conducted experiments on various datasets (Appendix C) to ensure the generalizability of our conclusions, as opposed to previous studies that relied on a few simple datasets. **2) Enhancing the understanding of the relationship between momentum and learning rate.** Leclerc & Madry (2020) presented a complex conclusion: momentum accelerates SGD under small learning rates but slows it down under large learning rates. This conclusion is inconsistent with previous findings (Yuan et al., 2016; Smith et al., 2020). The primary reason for this discrepancy is that SGDM experiences a larger effective learning rate $\frac{\eta}{1-\mu}$ compared to SGD's learning rate $\eta$ when their learning rates are the same. By considering the effective learning rate, we reconcile these differences and arrive at the following conclusions: a) Consistent with (Yuan et al., 2016), we find that SGDM and SGD have similar performance under small effective learning rates. b) Momentum accelerates SGD when the effective learning rate is greater than a specific threshold.

**Why** There are two mainstream explanations for the impact of momentum: **(E1)** Momentum can counteract the negative effects of high curvature. Smith et al. (2020) identified two regimes: "noise dominated" and "curvature dominated." They found that momentum is more effective in the "curvature dominated" regime. **(E2)** Momentum can cancel out the negative effects of stochastic noise, resulting in update directions that are more aligned with the gradient direction. However, both explanations have limitations:

- **(E1): Cannot explain the interplay between momentum and learning rate**. Smith et al. (2020) mainly categorized the two regimes based on batch size. This explanation cannot account for the diverse behavior of momentum with different learning rates under the same batch size.

- **(E2): Cannot explain why momentum accelerates GD**. If momentum's role is to cancel noise, then we would expect no effect of momentum in GD, where no noise exists. This is inconsistent with recent experiments (Kunstner et al., 2022; Kidambi et al., 2018; Shallue et al., 2019).

**Our work advances the understanding of momentum in the following ways**: 1) Our explanation can account for the interplay between momentum, learning rate, and batch size, and can also be applied to GD. 2) We establish a direct connection between the effect of momentum and the landscape (directional Hessian).

## 3    WHEN DOES MOMENTUM ACCELERATE SGD?

In this section, we explore under what circumstances momentum can accelerate SGD. In Section 3.1, we first establish a meaningful comparison framework for SGD and SGDM by considering the interplay between momentum and two factors, *i.e.* batch sizes and learning rates. In Section 3.2, we then conduct experiments under this framework and state our main observations.

### 3.1    A COMPARISON FRAMEWORK FOR SGDMS WITH DIFFERENT HYPERPARAMETERS

**Hyper-parameter scheduler.** We use constant step-size and constant momentum coefficient across the whole training process, *i.e.* , $\mu_t \equiv \mu$ and $\eta_t \equiv \eta$, as our primary objective is to understand the acceleration effect of SGDM rather than reproduce state-of-the-art performance.

**Effective learning rate.** Our aim is to study the essential influence of the momentum coefficient $\mu$ over the performance of SGDM. However, the momentum may affect the performance via different ways. For example, adding $\mu$ will change the update magnitude, which may have the same effect as changing the learning rate. Such effect can be approximated as follows,

$$\boldsymbol{m}_t = \sum_{s=1}^{t} \mu^{t-s} \nabla f_{B_s}(\boldsymbol{w}_s) \approx \frac{1-\mu^t}{1-\mu} \nabla f_{B_t}(\boldsymbol{w}_t) \rightarrow \frac{1}{1-\mu} \nabla f_{B_t}(\boldsymbol{w}_t) \text{ as } t \rightarrow \infty.$$

This indicates that SGDM with momentum coefficient $\mu$ and learning rate $\eta$ may have the same magnitude of update as SGD with learning rate $\frac{1}{1-\mu}\eta$. When comparing the performances of SGDM with different $\mu$, we want to exclude the effect of momentum that can be compensated by simply changing the learning rate. Therefore we introduce the concept of *effective learning rate* so that the different setup can be compared fairly to extract the essential effect of momentum.

Additionally, the batch size $b$ is another important hyperparameter in SGDM whose effect may be compensated by simply changing the learning rate. Specifically, we consider the gradient is averaged (rather than summed) over the individual samples in a minibatch. Larger batch size indicates fewer updates in one epoch. To compensate the number updates in one epoch, we adopt the the Linear Scaling Rule (Goyal et al., 2017) of the learning rate, which suggests that scaling the learning rate proportionally with the batch size keep the same convergence speed. This scaling rule has been verified to be effective for models and data with large sizes (Goyal et al., 2017).

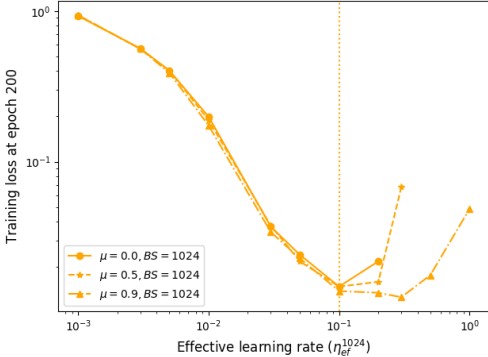 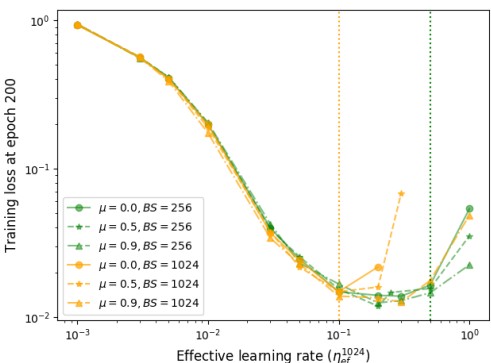

Figure 2: **The training speeds of SGD and SGDM exhibit an align-and-deviate pattern as the effective learning rate increases.** 1) When the effective learning rate is small, the training speeds of SGDM and SGD are almost the same. 2) After the effective learning rate beyond a certain threshold, SGDM outperforms SGD.

Figure 3: **The benefit of momentum is entangled with batch sizes** 1) Reducing batch size will defer the threshold point between align-and-deviation. 2)The larger momentum coefficient, the smaller gap of SGDMs with different batch sizes. 3) The gap between SGDM and SGD becomes larger when the batch size increases.

Putting these effects together, we propose to compare the performance between SGD and SGDM under the same *effective learning rate*, defined as follows:

$$
\eta_{\mathrm{ef}}^{k} = \underbrace{\frac{1}{1-\mu}}_{\substack{\text{Effect of} \\ \text{momentum}}} \cdot \overbrace{\frac{k}{b}}^{\substack{\text{Effect of} \\ \text{batch size}}} \cdot \; \eta,
$$

where $k$ is a reference batch size introduced for good visualization for typical choices of batch size and learning rate. **When there is no comparison across batch sizes, we simply choose $k = b$, and denote** $\eta_{\mathrm{ef}} = \eta_{\mathrm{ef}}^{b} = \frac{1}{1-\mu}\eta$.

**Rationale for Effective learning rate** The reason for defining an effective learning rate is to **maintain a consistent update magnitude across various parameter settings**. When adding momentum and keeping the learning rate fixed, the actual update magnitude increases. Regarding batch size, a larger batch size reduces the number of update iterations within an epoch. Previous studies (Goyal et al., 2017; Ma et al., 2018; Smith et al.) have compensated for this by increasing the learning rate.

**Measurement of performance.** As we care about the optimization performance of SGDM, we plot the training loss of SGDM after a prefixed number of epochs $T$, with respect to the effective learning rate for different settings of $\mu$ and $b$ (see Figure 2). We say one setting of SGDM outperforms another, if the former one has a smaller training loss after $T$ epochs for the same effective learning rate.

### 3.2 MOMENTUM ACCELERATES TRAINING ONLY FOR LARGE EFFECTIVE LEARNING RATES

We conduct experiments on the CIFAR10 dataset using VGG13-BN network[2]. We train SGDMs with batch size 1024 and three values of $\mu = \{0, 0.5, 0.9\}$, respectively, and choose $k = 1024$. We choose the epoch budget $T = 200$ (we show in Appendix C that our conclusion remains valid regardless of $T$). We note that our findings are also valid for other choices of batch sizes (see Section 3.3).

As discussed in Section 3.1, we plot curve of training losses with respect to effective learning rates for these three settings in Figure 2. We summarize the findings as follows.

---

[2]More experiments on different architectures and datasets are given in Appendix C. Our conclusion generally holds.

- **For small effective learning rates, SGDMs with different values of $\mu$ perform almost the same.** This indicates that momentum does not have the benefit of acceleration because one can always use SGD with a compensated learning rate to reach the performance of SGDM with a specific $\mu$.

- **As effective learning rate increases beyond some thresholds, the curves with small $\mu$ start deviating from the curve with large $\mu$ progressively, which implies some transition happens.** In this regime, we observe the benefit of momentum because simple compensation on learning rate does not helps SGDM with small $\mu$ reach the performance of SGDM with large $\mu$.

Overall, for the whole range of $\mu$, SGDM performs better than or equivalent to SGD. Such a neat relation can only be observed by introducing effective learning rates to align different values of $\mu$.

### 3.3 THE BENEFIT OF MOMENTUM IS ENTANGLED WITH BATCH SIZES

In this section, we examine the effect of batch size $b$ and understand how different batch sizes affect the benefit of momentum. We repeat the experiments in Section 3.2, i.e, experiments on the CIFAR10 dataset using VGG13-BN network for SGDM with 6 representative choices of hyperparameters $(\mu, b) \in \{0, 0.5, 0.9\} \times \{256, 1024\}$. We choose the epoch budget $T = 200$, the same as before. We plot the result of the training losses with respect to the effective learning rates in Figure 3.

Our findings are summarized as follows.

- **Batch size affects the deviation point.** Increasing the batch size, the SGD and SGDM will diverge at a smaller effective learning rate. This indicates that the acceleration of momentum interplays with learning rate and batch size.

- **The gap between SGDM and SGD becomes larger when the batch size increases.** This coincides with the observation that the acceleration effect of momentum is more pronounced with larger batch sizes (Kunstner et al., 2022; Kidambi et al., 2018; Shallue et al., 2019).

- **The larger momentum coefficient, the smaller gap of SGDMs with different batch sizes.** As one increases the momentum coefficient, the gap between the performances of SGDMs with different batch sizes becomes smaller.

## 4 WHY DOES MOMENTUM ACCELERATE SGD?

In Section 3, we have explored when momentum accelerates SGD. In this section, we want to understand why momentum accelerates SGD, or more precisely, the mechanism of momentum accelerating SGD. To control variable and simplify the analysis, we first focus on comparing GD and GDM, and find that deviation of GD and GDM is related with a phenomenon that *Hessian abruptly sharpens* along the update direction in Section 4.1. We then show that the abrupt sharpening of Hessian can explain the acceleration of momentum and batch size in Section 4.2 and 4.3, respectively.

**Remark:** The contribution of this section is in **explaining momentum acceleration through the lens of edge of stability**. The Propositions presented are not highly technical. The Papers (Lyu et al., 2022; Damian et al., 2022) give more detail information about the edge of stability phenomenon and cover most Propositions in this section.

### 4.1 HESSIAN ABRUPTLY SHARPENS WHEN GD DEVIATES FROM GDM

As the full-batch update is computationally expensive, we use a subset of CIFAR10 with 5K samples, which has been used to study GD behavior previously (Cohen et al., 2021; Ahn et al., 2022b). The network we use is fc-tanh(Cohen et al., 2021), i.e., a one-hidden-layer fully-connected network with 200 neurons and tanh activation.

We first verify that the align-and-deviate pattern still exist in this task in Figure 4A. We then pick one effective learning rate $\eta_{\mathrm{ef}} = 0.01$ before the deviation threshold and one effective learning rate $\eta_{\mathrm{ef}} = 0.1$ after the threshold and plot its training loss across epochs in Figure 4B and Figure 4C. We can see that for $\eta_{\mathrm{ef}} = 0.01$, the training curves of GDM and GD are smooth and closely aligned

(Figure 4B). When $\eta_{\text{ef}} = 0.1$ beyond the threshold in Figure 4A, the training curves of GDM and GD align with each other in first few epochs and then the loss of GD starts oscillating and deviates from the loss of GDM. It should be noted that GD becomes slower than GDM, i.e, the curve of GD is strictly on top of that of GDM, at the same moment that GD starts oscillating.

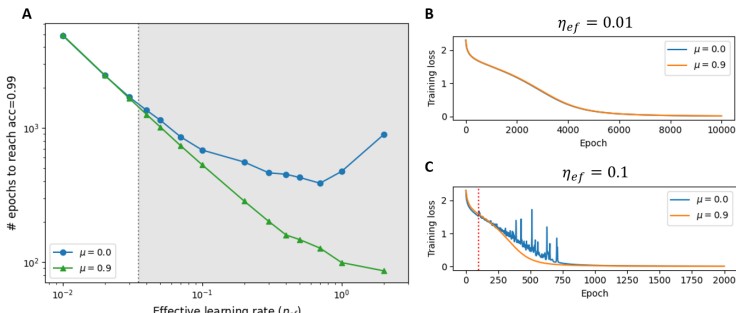

Figure 4: **Exploration of the training process on CIFAR10-5k dataset.** A: Experiments on Cifar10-5k gives a similar result as Figure 2. B: GD and GDM are aligned during the whole training process under small learning rate. C: GD and GDM are aligned before GD starting oscillating, and deviate after. The red dash line denotes the time when the GD starts oscillating.

The observation in Figure 4 provides a partial answer of why momentum accelerates GD by connecting it with preventing oscillation. However, we are still unclear the reason for why the oscillations happen and why GDM is less like to oscillate. With $\boldsymbol{w}_t$ be the iteration of GD, we revisit the Taylor expansion of the objective function $f$, which writes

$$f(\boldsymbol{w}_{t+1}) \approx f(\boldsymbol{w}_t) - \eta\|\nabla f(\boldsymbol{w}_t)\|^2 + \frac{\eta^2}{2}\nabla f(\boldsymbol{w}_t)^\top \nabla^2 f(\boldsymbol{w}_t)\nabla f(\boldsymbol{w}_t).$$

When the loss stably decreases, we have $f(\boldsymbol{w}_{t+1}) < f(\boldsymbol{w}_t)$, and based on the above approximation, we infer the *directional Hessian* along the update direction $H(\boldsymbol{w}_t, \boldsymbol{w}_{t+1} - \boldsymbol{w}_t) \stackrel{\triangle}{=} \frac{(\boldsymbol{w}_{t+1}-\boldsymbol{w}_t)^\top \nabla^2 f(\boldsymbol{w}_t)(\boldsymbol{w}_{t+1}-\boldsymbol{w}_t)}{\|\boldsymbol{w}_{t+1}-\boldsymbol{w}_t\|^2}$ satisfying $\eta H(\boldsymbol{w}_t, \boldsymbol{w}_{t+1} - \boldsymbol{w}_t) < 2$. On the other hand, we have $f(\boldsymbol{w}_{t+1}) \approx f(\boldsymbol{w}_t)$ when oscillating (Ahn et al., 2022b), and simple calculation gives $\eta H(\boldsymbol{w}_t, \boldsymbol{w}_{t+1} - \boldsymbol{w}_t) \approx 2$. Therefore, we conjecture that the oscillation is due to a sharp transition of the *directional Hessian* along the update.

To verify our conjecture, we plot the directional Hessian in Figure 5. We observe that there is a sharp transition of the directional Hessian along the update: it first stays around $0$ before oscillation, and then experiences a sudden jump at the time of oscillation. We referred to this phenomenon as "directional Hessian abrupt sharpening" or "abrupt sharpening" for short.

Abrupt sharpening explains why the oscillations happen. We further notice that abrupt sharpening is closely related to an existing concept called "edge of stability" (EoS), which describes the phenomenon that during the training of GD, sharpness, i.e., the maximum eigenvalue of Hessian, will gradually increase until it reaches $\frac{2}{\eta}$ and then hover at it. The phenomenon of gradually increasing sharpness is denoted as "progressive sharpening". It seems to contradict with abrupt sharpening of directional Hessian, but we show that abrupt sharpening is a joint outcome of progressive sharpening and renowned degenerate Hessian of deep neural networks (Sagun et al., 2016) through the following proposition. Consequently, abrupt sharpening can be viewed as a newfound component of EoS.

**Proposition 1.** *Given a minimization problem $\min_{\boldsymbol{w}\in\mathbb{R}^d} f(\boldsymbol{w})$, we consider minimizing its quadratic function approximation around a minimizer $\boldsymbol{w}^*$, i.e., $\tilde{f}(\boldsymbol{w}) \stackrel{\triangle}{=} \frac{1}{2}(\boldsymbol{w} - \boldsymbol{w}^*)^\top \nabla^2 f(\boldsymbol{w}^*)(\boldsymbol{w} - \boldsymbol{w}^*) + f(\boldsymbol{w}^*)$. Let $\boldsymbol{w}_t$ be the parameter given by GD with learning rate $\eta_{ef}$ at the $t$-th iteration. Let $\mathcal{A}$ be the space of eigenvectors of $\nabla^2 f(\boldsymbol{w}^*)$ corresponding to the maximum eigenvalue of $\boldsymbol{A}$. For almost every $\boldsymbol{w}_0 \in \mathbb{R}^d$, $\lim_{t\to\infty} \frac{\nabla \tilde{f}(\boldsymbol{w}_t)}{\|\nabla \tilde{f}(\boldsymbol{w}_t)\|} \in \mathcal{A}$ if and only if $\lambda_{\max}(\nabla^2 f(\boldsymbol{w}^*)) > \frac{2}{\eta_{ef}} - \lambda_{\min}(\nabla^2 f(\boldsymbol{w}^*))$.*

Sagun et al. (2016) observes that in deep learning tasks, the smallest eigenvalue of Hessian $\lambda_{\min}(\nabla^2 f(\boldsymbol{w}^*))$ is close to $0$. This together with Proposition 1 indicates that the update direction of GD would start to align with the eigenspace of the maximum eigenvalue only after the

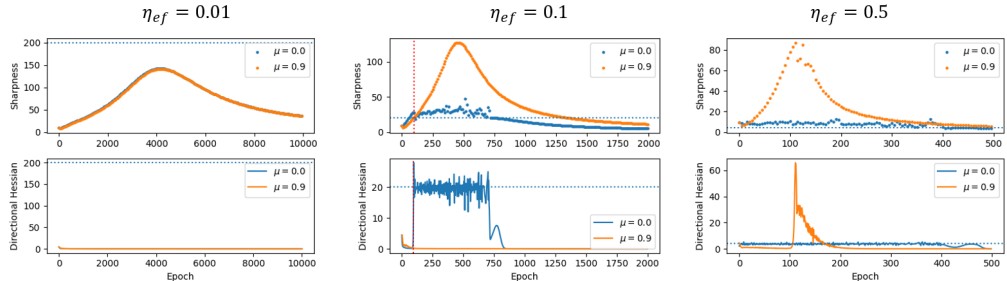

Figure 5: **Sharpness and directional Hessian on CIFAR10-5k dataset.** The dashed blue line represents the threshold $\frac{2}{\eta_{\text{ef}}}$. *Left:* With a small effective learning rate, the directional Hessian of GD and GDM are around $0$. *Center:* With a larger effective learning rate, GD exhibits abrupt sharpening and training loss starts to oscillate (marked by the red dash line) when the sharpness of GD surpasses $\frac{2}{\eta_{\text{ef}}}$, while directional Hessian of GDM stays around $0$. *Right:* With an even larger effective learning rate, both GDM and GD exhibits abrupt sharpening, but much later for GDM .

sharpness is very close to $\frac{2}{\eta}$ (thus in the early stage, directional sharpness stays around $0$). Once such an alignment starts, it is rapid because the convergence rate in Proposition 1 is exponential (please see the proof in Appendix E for details), which explains the abrupt sharpening of directional Hessian.

## 4.2 ABRUPT SHARPENING CAN EXPLAIN THE ACCELERATION OF MOMENTUM

Here we show that abrupt sharpening can be used to explain the acceleration of momentum.

**Small directional Hessian explains the alignment between GD and GDM.** Intuitively, when directional Hessian is relatively small, GD and GDM are like walking straightly on a line because small directional Hessian implies small change of gradient along the update direction. This agrees with the setting where we introduce effective learning rate, i.e., $\nabla f(\boldsymbol{w}_1) \approx \nabla f(\boldsymbol{w}_2) \approx \cdots \approx \nabla f(\boldsymbol{w}_t)$, and thus GDM performs similarly as GD under the same effective learning rate. This perfectly explains the alignment between GD and GDM before oscillation. We summarize the above intuition as the following property.

**Proposition 2.** *Denote the iterations of GD as $\{\boldsymbol{w}_t^{\text{GD}}\}_{t=1}^{\infty}$ and those of GDM as $\{\boldsymbol{w}_t^{\text{GDM}}\}_{t=1}^{\infty}$. If the directional Hessians satisfy $H(\boldsymbol{w}_s^{\text{GD}}, \boldsymbol{w}_{s+1}^{\text{GD}} - \boldsymbol{w}_s^{\text{GD}}) \approx 0$ and $H(\boldsymbol{w}_s^{\text{GDM}}, \boldsymbol{w}_{s+1}^{\text{GDM}} - \boldsymbol{w}_s^{\text{GDM}}) \approx 0$, $\forall s \leq t - 1$, then we have $f(\boldsymbol{w}_t^{\text{GD}}) \approx f(\boldsymbol{w}_t^{\text{GDM}})$.*

**Momentum defers abrupt sharpening, and thus accelerates GD.** First of all, we show again through quadratic programming that momentum has the effect to defer abrupt sharpening.

**Proposition 3.** *Let $f$, $\boldsymbol{w}^*$ and $\tilde{f}$ and $\mathcal{A}$ be defined in Proposition 3. Let $\boldsymbol{w}_t$ be the parameter given by GDM at the $t$-th iteration. Then, for almost everywhere $\boldsymbol{w}_0 \in \mathbb{R}^d$, $\lim_{t \to \infty} \frac{\nabla \tilde{f}(\boldsymbol{w}_t)}{\|\nabla \tilde{f}(\boldsymbol{w}_t)\|} \in \mathcal{A}$ if and only if $\lambda_{\max}(\nabla^2 f(\boldsymbol{w})) > \frac{2(1+\mu)}{(1-\mu)\eta_{\text{ef}}} - \lambda_{\min}(\nabla^2 f(\boldsymbol{w}))$.*

Comparing Proposition 3 with Proposition 1, we observe that with a relative small $\lambda_{\min}$, the required $\lambda_{\max}$ for abrupt sharpening appearance of GDM is $\frac{(1+\mu)}{1-\mu}$ (which is 19 when $\mu = 0.9$) times larger than that of GD. As the sharpness progressively increases, reaching the required sharpness of GDM takes a much longer time than reaching that of GD (an extreme case is that abrupt sharpening happens in GD but not in GDM). Meanwhile, entrance of edge of stability has been known to slow down the convergence. In Ahn et al. (2022a), it is shown that when not entering EoS, GD converges in $\mathcal{O}(1/\eta_{\text{ef}})$ iterations, but require $\Omega(1/\eta_{\text{ef}}^2)$ iterations to converge in the EoS regime. Together, we arrive at the conclusion that momentum can accelerate GD via deferring the entrance of EoS (abrupt sharpening).

### 4.3 Extending the analysis to stochastic case: interplay between momentum and batch size

Over the same experiment of Sections 4.1 and 4.2, we first plot the training curves of GD, GDM, and SGD with batch size 250 and $\eta_{\text{ef}}^{5000} = 0.1$ in Figure 6A. Specifically, we find that **stochastic noise can also defer abrupt sharpening**: GD enters EoS during the training process, while SGD and GDM does not and they remain well-aligned throughout the training process.

Since in previous section, we have explained that entrance of EoS slows down the convergence, such an observation explains why in Figure 2, reducing batch size also accelerates SGDM with respect to the number of passes of the data. Furthermore, this observation also explains why the effect of momentum is more pronounced when batch size is large since stochastic noise and momentum has an overlapping effect in preventing abrupt sharpening.

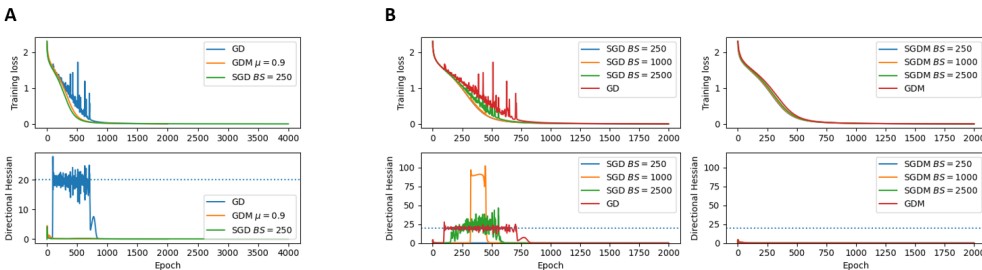

Figure 6: **Reducing batch size and adding momentum play a similar role in preventing abrupt sharpening.** A: Reducing batch size can help prevent abrupt sharpening. B: Adding momentum can extend the range of batch sizes where linear scaling rule holds.

From Figure 6B we can see that the performance of large-batch SGD is worse than small-batch SGD because large-batch enters EoS while small-batch does not. When momentum is added, large-batch also does not enter EoS.

**Why batch size impacts the deviation point**   The deviation happens when the SGD enters EoS while SGDM is not enter the EoS. After decreasing the batch size, larger learning rate is required for SGD to enter EoS. Therefore, the deviation point is defered.

## 5 Conclusion

This paper investigates the relationship between momentum, learning rate, and batch size. We observe an align-and-deviate pattern when either fixing the batch size and increasing the effective learning rate (Figure 2) or fixing the effective learning rate and increasing the batch size (Figure 13). Before the deviation point, the training speed of SGD and SGDM are almost the same. However, after the deviation point, SGDM outperforms SGD. We link the phase transition to the EoS and explain that momentum accelerates training via preventing the entrance of EoS. We also observe and analyze the effect of batch size following the above framework. In summary, this paper provides thorough empirical result to see and analyze when and why momentum accelerates SGD under various settings.

## 6 Limitation

Our current paper has two limitations: **1) Inability to explain situations where $\mu$ is close to 1.** In Appendix D, we analyze that leveraging extremely large $\mu$ can actually worsen the acceleration of momentum. However, since this setting is rarely used in practice, it is also overlooked by most previous works (Defazio, 2020; Cutkosky & Mehta, 2020; Polyak, 1964; 1987; Yuan et al., 2016; Leclerc & Madry, 2020; Smith et al., 2020). **2) The evaluation of model architectures and datasets is not comprehensive.** In Appendix C, we conduct experiments on commonly used model architectures and popular datasets. Our findings are based on these configurations. Although it is impossible to test all scenarios, more extensive experiments are needed in future work.

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

## A  OTHER RELATED WORK

**Edge of Stability.**  (Cohen et al., 2021) discovers a negative correlation between the sharpness of objective functions in the training process of deep learning tasks and the learning rate, called "Edge of Stability" (EoS). Specifically, when using gradient descent (GD) with learning rate $\eta$, they observe that the sharpness will first progressively increase, and then hover at $\frac{2}{\eta}$. Similar phenomena are latter observed in other optimizers including SGD, SGDM, and Adam (Cohen et al., 2022). Traditionally, optimization analysis requires sharpness to be smaller than $\frac{2}{\eta}$ to ensure convergence. This is, however, violated by EoS, and several works have tried to understand such a mismatch theoretically. Interesting readers can refer to (Ma et al., 2022; Ahn et al., 2022b; Arora et al., 2022; Li et al., 2022; Ahn et al., 2022a; Zhu et al., 2022) for details.

**Linear scaling rule of learning rate with batch size.** Linear scaling rule is first proposed by (Goyal et al., 2017), suggesting that the when the batch size is smaller than a certain threshold (called critical batch size), scaling the learning rate according to the batch size keep the performance the same. Such a law is further theoretically verified by (Ma et al., 2018) which study SGD over quadratic functions. (Ma et al., 2018) also show that using linear scaling law above the critical batch size hurt the performance, which is empirically observed by (Chen et al., 2018). These methodologies are used in (Smith et al.) to decay learning rate.

## B  FURTHER EXPERIMENTS ON OTHER OPTIMIZERS

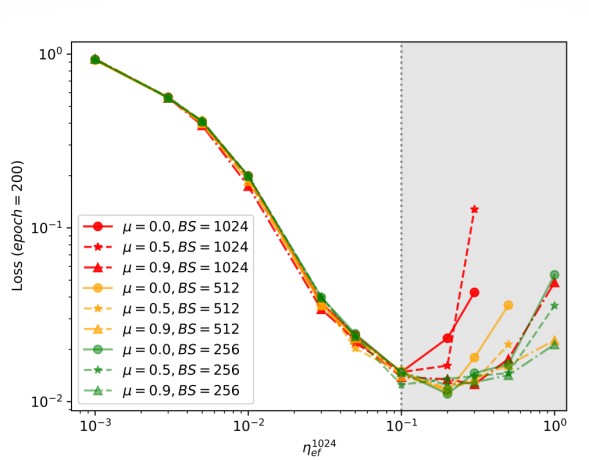

Figure 7: **The align-and-deviate pattern for Nesterov's Momentum.** The behavior of Nesterov's momentum is similar to that of Polyak's momentum (see Figure 3) when increasing the effective learning rate.

### B.1  ON THE EFFECT OF NESTEROV'S MOMENTUM

To give a full picture of the effect of momentum, we further conduct experiments over SGD with Nesterov's momentum as a complement to the discussion about the Polyak's momentum in the main text. Specifically, the update rule of Nesterov's momentum is given in Algorithm 2, which is the implementation in PyTorch.

---

**Algorithm 2** SGD with Nesterov's Momentum

---

1: **Input:** the loss function $\ell(w, z)$, the initial point $\boldsymbol{w}_1 \in \mathbb{R}^d$, the batch size $b$, learning rates $\{\eta_t\}_{t=1}^T$, $\boldsymbol{m}_0 = 0$, and momentum hyperparameters $\{\mu_t\}_{t=1}^T$.
2: **For** $t = 1 \rightarrow T$:
3:    Sample a mini-batch of data $B_t$ with size $b$
4:    Calculate stochastic gradient $\nabla f_{B_t}(w_t) = \frac{1}{b} \sum_{z \in B_t} \ell(w_t, z)$
5:    Update $\boldsymbol{m}_t \leftarrow \mu_t \boldsymbol{m}_{t-1} + \nabla f_{B_t}(\boldsymbol{w}_t)$
6:    Update $\boldsymbol{w}_{t+1} \leftarrow \boldsymbol{w}_t - \eta_t(\mu_t \boldsymbol{m}_t + \nabla f_{B_t}(\boldsymbol{w}_t))$
7: **End For**

---

#### B.1.1  DERIVATION OF EFFECTIVE LEARNING RATE

Like in the analysis of Polyak's momentum, we fix $\eta_t$ and $\mu_t$ to be constants. We show below that Nesterov's momentum has the similar effect as Polyak's momentum to amplify the update magnitude. Specifically, we have

$$\boldsymbol{m}_t = \sum_{s=1}^t \mu^{t-s} \nabla f_{B_s}(\boldsymbol{w}_s) \approx \frac{1 - \mu^t}{1 - \mu} \nabla f_{B_t}(\boldsymbol{w}_t) \rightarrow \frac{1}{1 - \mu} \nabla f_{B_t}(\boldsymbol{w}_t) \text{ as } t \rightarrow \infty,$$

and thus

$$\mu \boldsymbol{m}_t + \nabla f_{B_t}(\boldsymbol{w}_t) \approx \frac{1}{1 - \mu} \nabla f_{B_t}(\boldsymbol{w}_t) \text{ as } t \to \infty.$$

To rule out such a effect, we define the effective learning rate of Nesterov's momentum as

$$\eta_{\text{ef}}^k = \frac{1}{1 - \mu} \cdot \frac{k}{b} \cdot \eta.$$

### B.1.2 EXPERIMENTS

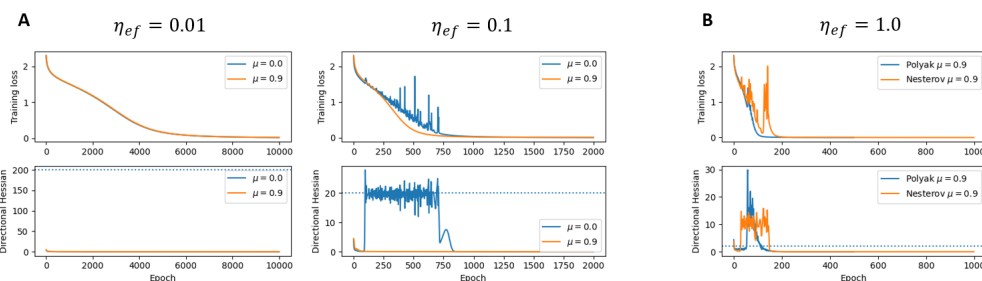

Figure 8: **Exploration of Nesterov's momentum** A: Nesterov's can also prevent the abrupt sharpening. B: Compared with Polyak, Nesterov's performs worse in preventing abrupt sharpening. Nesterov's GDM enters EoS earlier than Polyak's momentum. Additionally, the training speed of Nesterov's momentum is slower.

We conduct the experiments of SGD with Nesterov's momentum under the same setup as Figure 3. We plot the results in Figure 7. We can see that optimizers with Nesterov's momentum behave similarly to the counterparts with Polyak's momentum as shown in Figure 3. Furthermore, we provide a further investigation on Nesterov's momentum by conducting an experiment based on the setup of Figure 4, plotted in Figure 8. Figure 8A shows that the Nesterov momentum can also prevent abrupt sharpening during the training process. Then, we give a simple comparison between Polyak's and Nesterov's momentum by comparing them together under $\eta_{\text{ef}} = 1.0$, where both of them will enter the EoS. In this setting, we find that compared with SGD with Polyak's momentum, SGD with Nesterov's momentum with same $\mu$ entere EoS earlier (Figure 8B), and Polyak's momentum performs better than Nesterov's momentum under this setting. However, future work with more extensive experiments is required before making any conclusive claim on optimizers with Nesterov's momentum. In this paper, we focus on the optimizers with Polyak's momentum.

### B.2 ON THE EFFECT OF MOMENTUM IN ADAM

---

**Algorithm 3** Adam

---

1: **Input:** the loss function $\ell(w, z)$, the initial point $\boldsymbol{w}_1 \in \mathbb{R}^d$, the batch size $b$, learning rates $\{\eta_t\}_{t=1}^T$, $\boldsymbol{m}_0 = 0, \boldsymbol{v} = 0$, and hyperparameters $\beta = (\beta_1, \beta_2)$.
2: **For** $t = 1 \to T$:
3:     Sample a mini-batch of data $B_t$ with size $b$
4:     Calculate stochastic gradient $\nabla f_{B_t}(w_t) = \frac{1}{b} \sum_{z \in B_t} \ell(w_t, z)$
5:     Update $\boldsymbol{m}_t \leftarrow \beta_1 \boldsymbol{m}_{t-1} + (1 - \beta_1) \nabla f_{B_t}(\boldsymbol{w}_t)$
6:     Update $\boldsymbol{v}_t \leftarrow \beta_2 \boldsymbol{v}_{t-1} + (1 - \beta_2) \nabla f_{B_t}(\boldsymbol{w}_t)^{\odot 2}$
7:     Update $\boldsymbol{w}_{t+1} \leftarrow \boldsymbol{w}_t - \eta_t \frac{\boldsymbol{m}_t/(1-\beta_1^t)}{\sqrt{\boldsymbol{v}_t/(1-\beta_2^t)} + \epsilon}$
8: **End For**

---

Here we step beyond SGD and provide a preliminary investigation on the effect of momentum in Adam (Kingma & Ba, 2014). The psedocode of Adam is given in Algorithm 3. We first derive the effective learning rate of Adam. Since

$$\boldsymbol{m}_t = (1 - \beta_1) \sum_{s=1}^t \beta_1^{t-s} \nabla f_{B_s}(\boldsymbol{w}_s) \approx (1 - \beta_1^t) \nabla f_{B_t}(\boldsymbol{w}_t) \to \nabla f_{B_t}(\boldsymbol{w}_t) \text{ as } t \to \infty,$$

we define the effective learning rate of Adam directly as the learning rate *i.e.* $\eta_{\text{ef}} = \eta$ (here we do not consider the effect of batch size since it is still an open problem for the effect of batch size in Adam). We conduct the experiments of full-batch Adam under the same setup as Figure 4. Since we focus on the effect of momentum, we fix $\beta_2 = 0.999$ (which is the default value in PyTorch) and choose $\beta_1$ from $\{0, 0.5, 0.9\}$. The results are plotted in Figure 9.

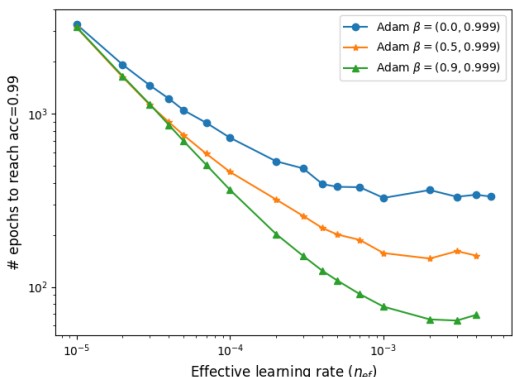

Figure 9: **The align-and-deviate pattern also exists in Adam.** When increasing the effective learning rate, Adam with different $\beta_1$ also exhibits an align-and-deviate pattern. Here $\beta = (\beta_1, \beta_2)$.

## C  MORE EXPLORATION ON THE ALIGN-AND-DEVIATE PATTERN

**Influence of model architecture.** In this study, we investigate whether varying model designs have an impact on the final conclusions. We set the batch size to 1024 and allocate an epoch budget of T = 200. The experiments are carried out using the Cifar10 dataset. All the considered architectures exhibit the align-and-deviate pattern. However, the effect of momentum varies across different models. For instance, momentum plays a more significant role in improving performance for VGG13 than that for VGG13BN as observed in Figure 10. Moreover, momentum is particularly important for training the ViT(Dosovitskiy et al., 2020) model, as depicted in Figure 10 (ViT).

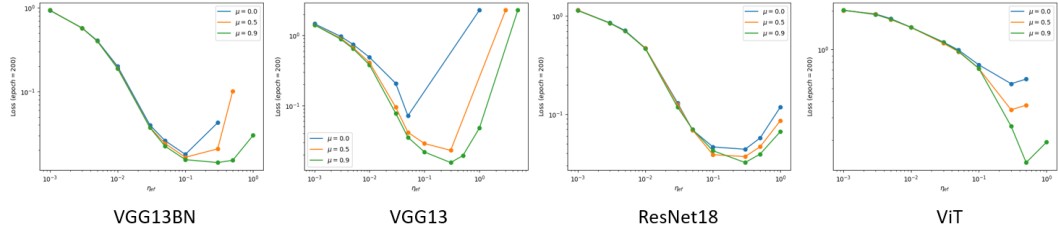

Figure 10: **Experiments with different neural network architectures.** Momentum has a more significant role in VGG13 and ViT network compared with ResNet18(He et al., 2016) and VGG13BN network.

**Influence of datasets** The experiments, as illustrated in Figure 11, are carried out using a variety of datasets, such as Cifar100(Simonyan & Zisserman, 2014), WikiText2(Merity et al., 2016), and ImageNet(Deng et al., 2009). For each dataset, we employ a different model: VGG13BN for Cifar100, Transformer[3] for WikiText2, and ResNet18 for ImageNet. We consistently observe the align-and-deviate pattern across these datasets. However, the positions of deviation points differ considerably among them. This variation could be attributed to factors such as dataset size, task difficulty, and other aspects.

---

[3]https://pytorch.org/tutorials/beginner/transformer_tutorial.html

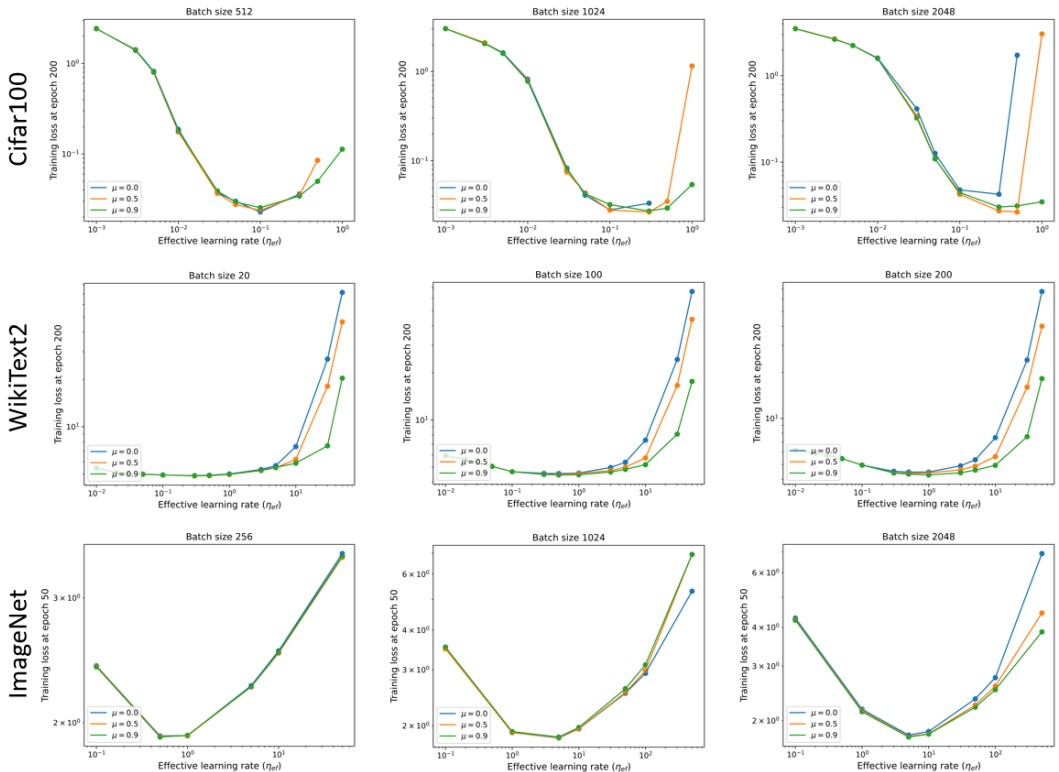

Figure 11: **Experiments across different datasets.** The align-and-deviate pattern is consistently observed along these datasets.

**Different epoch setting.** In this paper, we use the training loss at epoch $T$ to represent the training speed of optimizers. The $T$ is chosen to be 200 in our experiments. Here, we explore different values of T from $\{50, 100, 150, 200\}$, and we want to check whether the choice of $T$ matters. From Figure 12, we observe that the align-and-deviate pattern exists no matter what value of $T$ is chosen.

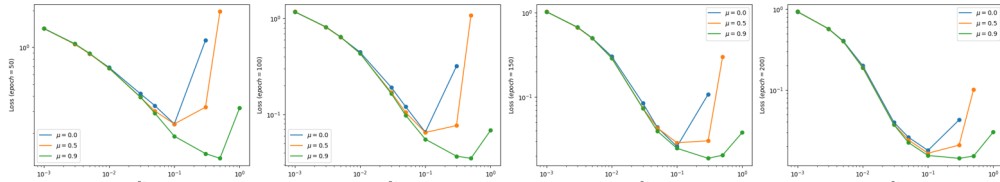

Figure 12: **Exploration the align-and-deviation pattern with different epoch settings.** The align-and-deviate pattern is observed in all these settings.

**Influence of batchsize** We conduct additional experiments to explore the imfluence of batchsize. Specifically, we fix the effective learning rate $\eta_{\text{ef}}^{1024} = 0.1$ and gradually increase the batch size to plot a curve of training loss with respect to the batch size in Figure 13. We observe that when batch sizes are small, SGDM with different $\mu$s performs almost the same, and when the batch size increases beyond a threshold, SGDM with larger $\mu$ tends to perform better.

Moreover, we note that a horizontal curve in Figure 13 is equivalent to the Linear Scaling Law (Goyal et al., 2017) and we can see that **momentum extends the range of batch sizes in which the Linear Scaling Law holds.**

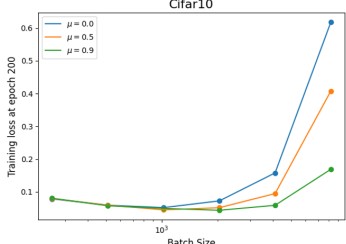 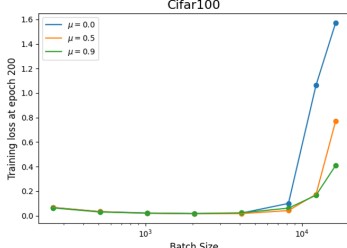

Figure 13: **Momentum extends the range of batch sizes in which the Linear Scaling Law holds.** 1) A similar align-and-deviate pattern for SGD and SGDM is also discovered when the batch size is increased. 2) Each curve remains nearly horizontal up to a specific threshold batch size. The threshold batch size for SGDM is greater than that for SGD.

# D  SITUATION WHEN MOMENTUM COEFFICIENT IS CLOSE TO 1

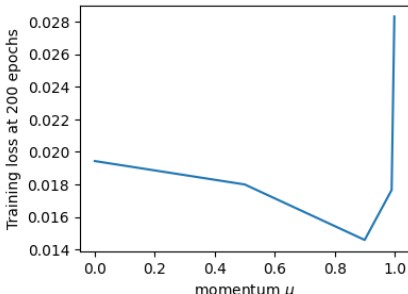

Figure 14: Exploration on the case when $\mu$ is close to 1. The experiments are conducted on Cifar10 using VGG13-BN with batch size 1024.

- Momentum will slow-down the SGD when $\mu \to 1$. Figure 14 explore the behavior of SGDM under different $\mu$. The performance of SGDM will suddenly decrease when the $\mu$ is larger than 0.9.

# E  PROOFS OF THEORETICAL RESULTS

## E.1  PROOF OF PROPOSITIONS 1 AND 3

*Proof of Proposition 1.* To begin with, by linear transformation, we can assume without loss of generality that $A$ is a diagonal matrix, $b = 0$ and $c = 0$. Denote $A = \text{Diag}(\lambda_1, \cdots, \lambda_d)$, where $\lambda_{\max}(A) = \lambda_1 \geq \lambda_2 \geq \cdots \geq \lambda_d = \lambda_{\min}(A)$. Denote $w_t = (w_{t,1}, \cdots, w_{t,d})$. Let $m_1$ be the number of eigenvalues equal to $\lambda_{\max}$. Let $m_2$ be the number of eigenvalues equal to $\lambda_{\min}$. Then, $\mathcal{A} = \text{span}\{e_1, \cdots, e_{m_1}\}$. Based on the update rule of GD, we obtain that

$$w_{t+1,i} = (1 - \eta\lambda_i)w_{t,i},$$

and thus $w_{t,i} = (1 - \eta\lambda_i)^t w_{0,i}$. Let $\mathcal{K} = \{x : x_{d-m_2+1} = \cdots = x_d = 0\} \cup \{x : x_1 = \cdots = x_{m_1} = 0\}$. Obviously, $\mathcal{K}$ is a zero-measure set. Then, we have that $\lim_{t\to\infty} \frac{w_t}{\|w_t\|} \in \mathcal{A}$ if and only if $|1 - \eta\lambda_1| > |1 - \eta\lambda_d|$, which gives $\eta > \frac{2}{\lambda_1 + \lambda_d}$.  □

*Proof of Proposition 3.* Let $\lambda_1, \cdots, \lambda_d$, $\mathcal{K}$, $m_1$, and $m_2$ be defined in the proof of Proposition 1. Then, the update rule of GDM gives

$$w_{t+1,i} = w_{t,i} + \mu(w_{t,i} - w_{t-1,i}) - (1 - \mu)\eta_{\text{ef}}\partial_i f(w_{t,i})$$
$$= w_{t,i} + \mu(w_{t,i} - w_{t-1,i}) - (1 - \mu)\eta_{\text{ef}}\lambda_i w_{t,i}.$$

Solving the above series gives

$$\boldsymbol{w}_{t,i} = c_{i,1}d_{i,1}^t + c_{i,2}d_{i,2}^t,$$

where $d_{i,1} = \frac{(1+\mu)-(1-\mu)\eta_{\text{ef}}\lambda_i}{2} + \sqrt{\left(\frac{(1+\mu)-(1-\mu)\eta_{\text{ef}}\lambda_i}{2}\right)^2 - \mu}$, and $d_{i,2} = \frac{(1+\mu)-(1-\mu)\eta_{\text{ef}}\lambda_i}{2} - \sqrt{\left(\frac{(1+\mu)-(1-\mu)\eta_{\text{ef}}\lambda_i}{2}\right)^2 - \mu}$.

Therefore, $\lim_{t\to\infty}\frac{\boldsymbol{w}_t}{\|\boldsymbol{w}_t\|} \in \mathcal{A}$ if and only if $\max\{d_{1,1}, d_{1,2}\} > \max_{i\neq 1}\{d_{i,1}, d_{i,2}\}$. On the other hand, note that $g(x) = \max\{|\frac{(1+\mu)-x}{2} + \sqrt{\left(\frac{(1+\mu)-x}{2}\right)^2 - \mu}|, |\frac{(1+\mu)-x}{2} - \sqrt{\left(\frac{(1+\mu)-x}{2}\right)^2 - \mu}|\}$ is symmetric with respect to $x = 1 + \mu$, and the maximum value of $g(x)$ over any interval $[a, b]$ is achieved at $a$ or $b$, then $\lim_{t\to\infty}\frac{\boldsymbol{w}_t}{\|\boldsymbol{w}_t\|} \in \mathcal{A}$ if and only if $(1-\mu)\eta_{\text{ef}}\lambda_1 + (1-\mu)\eta_{\text{ef}}\lambda_d > 2(1+\mu)$ and $(1-\mu)\eta_{\text{ef}}\lambda_1 > (1+\sqrt{\mu})^2$.

The proof is completed. $\qquad\square$

### E.2 PROOF OF PROPOSITION 2

Without loss of generality, choose $\boldsymbol{m}_0 = \frac{\nabla f(\boldsymbol{w}_1^{\text{GDM}})}{1-\mu}$ (since the influence of $\boldsymbol{m}_0$ diminishes exponentially fast). To begin with, define an auxiliary sequence as $\boldsymbol{u}_1 = \boldsymbol{w}_1^{\text{GDM}} - \frac{\mu}{1-\mu}\eta_{\text{ef}}\nabla f(\boldsymbol{w}_1^{\text{GDM}})$ and $\boldsymbol{u}_t = \frac{\boldsymbol{w}_t^{\text{GDM}}-\mu\boldsymbol{w}_{t-1}^{\text{GDM}}}{1-\mu}$. One can easily verify that the update rule of GDM is equivalent to

$$\boldsymbol{u}_{t+1} = \boldsymbol{u}_t - \eta_{\text{ef}}\nabla f(\boldsymbol{w}_t^{\text{GDM}}), \boldsymbol{w}_{t+1} = (1-\mu)\boldsymbol{u}_{t+1} + \mu\boldsymbol{w}_t^{\text{GDM}}. \tag{1}$$

When $t = 1$, we have

$$\|\nabla f(\boldsymbol{w}^{\text{GDM}})\|^2 = \|\nabla f(\boldsymbol{w}^{\text{GD}})\|^2$$

by definition. We then show that when $t \geq k \geq 2$, $f(\boldsymbol{u}_k) - f(\boldsymbol{u}_{k-1}) \approx f(\boldsymbol{w}_k^{\text{GD}}) - f(\boldsymbol{w}_{k-1}^{\text{GD}})$ and $\|\nabla f(\boldsymbol{w}_k^{\text{GD}})\| \approx \|\nabla f(\boldsymbol{w}_k^{\text{GDM}})\|$ by induction. Suppose that the claim holds for the $k$-th iteration. Then, for the $(k + 1)$-th iteration, by Taylor's expansion, we have

$$f(\boldsymbol{u}_k) \approx f(\boldsymbol{w}_{k-1}^{\text{GDM}}) + \langle \boldsymbol{u}_k - \boldsymbol{w}_{k-1}^{\text{GDM}}, \nabla f(\boldsymbol{w}_{k-1}^{\text{GDM}})\rangle + \frac{H(\boldsymbol{w}_{k-1}^{\text{GDM}}, \boldsymbol{u}_k - \boldsymbol{w}_{k-1}^{\text{GDM}})}{2}\|\boldsymbol{u}_k - \boldsymbol{w}_{k-1}^{\text{GDM}}\|^2,$$

$$f(\boldsymbol{u}_{k+1}) \approx f(\boldsymbol{w}_k^{\text{GDM}}) + \langle \boldsymbol{u}_{k+1} - \boldsymbol{w}_k^{\text{GDM}}, \nabla f(\boldsymbol{w}_k^{\text{GDM}})\rangle + \frac{H(\boldsymbol{w}_k^{\text{GDM}}, \boldsymbol{u}_{k+1} - \boldsymbol{w}_k^{\text{GDM}})}{2}\|\boldsymbol{u}_{k+1} - \boldsymbol{w}_k^{\text{GDM}}\|^2,$$

$$f(\boldsymbol{w}_k^{\text{GDM}}) \approx f(\boldsymbol{w}_{k-1}^{\text{GDM}}) + \langle \boldsymbol{w}_k^{\text{GDM}} - \boldsymbol{w}_{k-1}^{\text{GDM}}, \nabla f(\boldsymbol{w}_{k-1}^{\text{GDM}})\rangle + \frac{H(\boldsymbol{w}_{k-1}^{\text{GDM}}, \boldsymbol{w}_k^{\text{GDM}} - \boldsymbol{w}_{k-1}^{\text{GDM}})}{2}\|\boldsymbol{w}_k^{\text{GDM}} - \boldsymbol{w}_{k-1}^{\text{GDM}}\|^2.$$

By the definition of $\boldsymbol{u}_k$, we have that $\boldsymbol{u}_k - \boldsymbol{w}_{k-1}^{\text{GDM}} = \frac{\boldsymbol{w}_k^{\text{GDM}}-\boldsymbol{w}_{k-1}^{\text{GDM}}}{1-\mu}$, and thus $H(\boldsymbol{w}_{k-1}^{\text{GDM}}, \boldsymbol{u}_k - \boldsymbol{w}_{k-1}^{\text{GDM}}) = H(\boldsymbol{w}_{k-1}^{\text{GDM}}, \boldsymbol{w}_k^{\text{GDM}} - \boldsymbol{w}_{k-1}^{\text{GDM}}) \approx 0$. Similarly, we have $H(\boldsymbol{w}_k^{\text{GDM}}, \boldsymbol{u}_{k+1} - \boldsymbol{w}_k^{\text{GDM}}) \approx 0$ and $H(\boldsymbol{w}_{k-1}^{\text{GDM}}, \boldsymbol{w}_k^{\text{GDM}} - \boldsymbol{w}_{k-1}^{\text{GDM}}) \approx 0$. Therefore, summing up the above three equations, we have

$$f(\boldsymbol{u}_{k+1}) \approx f(\boldsymbol{u}_k) + \langle \boldsymbol{u}_{k+1} - \boldsymbol{w}_k^{\text{GDM}}, \nabla f(\boldsymbol{w}_k^{\text{GDM}})\rangle + \langle \boldsymbol{w}_k^{\text{GDM}} - \boldsymbol{u}_k, \nabla f(\boldsymbol{w}_{k-1}^{\text{GDM}})\rangle.$$

Since

$$\langle \boldsymbol{w}_k^{\text{GDM}} - \boldsymbol{w}_{k-1}^{\text{GDM}}, \nabla f(\boldsymbol{w}_{k-1}^{\text{GDM}})\rangle$$
$$=\langle \boldsymbol{w}_k^{\text{GDM}} - \boldsymbol{w}_{k-1}^{\text{GDM}}, \nabla f(\boldsymbol{w}_k^{\text{GDM}})\rangle - \langle \boldsymbol{w}_k^{\text{GDM}} - \boldsymbol{w}_{k-1}^{\text{GDM}}, \nabla f(\boldsymbol{w}_k^{\text{GDM}}) - \nabla f(\boldsymbol{w}_{k-1}^{\text{GDM}})\rangle$$
$$\approx\langle \boldsymbol{w}_k^{\text{GDM}} - \boldsymbol{w}_{k-1}^{\text{GDM}}, \nabla f(\boldsymbol{w}_k^{\text{GDM}})\rangle - H(\boldsymbol{w}_{k-1}^{\text{GDM}}, \boldsymbol{w}_k^{\text{GDM}} - \boldsymbol{w}_{k-1}^{\text{GDM}})\|\boldsymbol{w}_k^{\text{GDM}} - \boldsymbol{w}_{k-1}^{\text{GDM}}\|^2$$
$$\approx\langle \boldsymbol{w}_k^{\text{GDM}} - \boldsymbol{w}_{k-1}^{\text{GDM}}, \nabla f(\boldsymbol{w}_k^{\text{GDM}})\rangle,$$

we further have

$$f(\boldsymbol{u}_{k+1}) \approx f(\boldsymbol{u}_k) + \langle \boldsymbol{u}_{k+1} - \boldsymbol{w}_k^{\text{GDM}}, \nabla f(\boldsymbol{w}_k^{\text{GDM}})\rangle + \langle \boldsymbol{w}_k^{\text{GDM}} - \boldsymbol{u}_k, \nabla f(\boldsymbol{w}_k^{\text{GDM}})\rangle$$
$$= f(\boldsymbol{u}_k) + \langle \boldsymbol{u}_{k+1} - \boldsymbol{u}_k^{\text{GDM}}, \nabla f(\boldsymbol{w}_k^{\text{GDM}})\rangle$$
$$= f(\boldsymbol{u}_k) - \eta_{\text{ef}}\|\nabla f(\boldsymbol{w}_k^{\text{GDM}})\|^2.$$

Following the same routine, we obtain
$$f(\boldsymbol{w}_{k+1}^{\mathrm{GD}}) \approx f(\boldsymbol{w}_k^{\mathrm{GD}}) - \eta_{\mathrm{ef}}\|\nabla f(\boldsymbol{w}_k^{\mathrm{GD}})\|^2,$$
and thus we obtain $f(\boldsymbol{u}_{k+1}) - f(\boldsymbol{u}_k) \approx f(\boldsymbol{w}_{k+1}^{\mathrm{GD}}) - f(\boldsymbol{w}_k^{\mathrm{GD}})$ due to that $\|\nabla f(\boldsymbol{w}_k^{\mathrm{GD}})\|^2 \approx \|\nabla f(\boldsymbol{w}_k^{\mathrm{GDM}})\|^2$ by the induction hypothesis.

Meanwhile, we have
$$
\begin{aligned}
\|\nabla f(\boldsymbol{w}_{k+1}^{\mathrm{GD}})\|^2 &\approx \|\nabla f(\boldsymbol{w}_k^{\mathrm{GD}})\|^2 + \langle \nabla f(\boldsymbol{w}_k^{\mathrm{GD}}), \nabla f(\boldsymbol{w}_{k+1}^{\mathrm{GD}}) - \nabla f(\boldsymbol{w}_k^{\mathrm{GD}}) \rangle \\
&\approx \|\nabla f(\boldsymbol{w}_k^{\mathrm{GD}})\|^2 + \langle \nabla f(\boldsymbol{w}_k^{\mathrm{GD}}), \nabla^2 f(\boldsymbol{w}_k^{\mathrm{GD}})(\nabla f(\boldsymbol{w}_{k+1}^{\mathrm{GD}}) - \nabla f(\boldsymbol{w}_k^{\mathrm{GD}})) \rangle \\
&= \|\nabla f(\boldsymbol{w}_k^{\mathrm{GD}})\|^2 + \eta_{\mathrm{ef}} H(\boldsymbol{w}_k^{\mathrm{GD}}, \boldsymbol{w}_{k+1}^{\mathrm{GD}} - \boldsymbol{w}_k^{\mathrm{GD}})\|\nabla f(\boldsymbol{w}_{k+1}^{\mathrm{GD}})\|^2 \approx \|\nabla f(\boldsymbol{w}_k^{\mathrm{GD}})\|^2.
\end{aligned}
$$
Following the similar routine, we obtain
$$\|\nabla f(\boldsymbol{w}_{k+1}^{\mathrm{GDM}})\|^2 \approx \|\nabla f(\boldsymbol{w}_k^{\mathrm{GDM}})\|^2,$$
and thus we obtain $\|\nabla f(\boldsymbol{w}_{k+1}^{\mathrm{GD}})\|^2 \approx \|\nabla f(\boldsymbol{w}_{k+1}^{\mathrm{GDM}})\|^2$ due to that $\|\nabla f(\boldsymbol{w}_k^{\mathrm{GD}})\|^2 \approx \|\nabla f(\boldsymbol{w}_k^{\mathrm{GDM}})\|^2$ by the induction hypothesis.

As a conclusion, we obtain that $f(\boldsymbol{u}_t) - f(\boldsymbol{u}_1) \approx f(\boldsymbol{w}_t^{\mathrm{GD}}) - f(\boldsymbol{w}_1^{\mathrm{GD}})$.

Meanwhile, as discussed above, we have
$$
\begin{aligned}
f(\boldsymbol{u}_t) &\approx f(\boldsymbol{w}_{k-1}^{\mathrm{GDM}}) + \langle \boldsymbol{u}_t - \boldsymbol{w}_{t-1}^{\mathrm{GDM}}, \nabla f(\boldsymbol{w}_{t-1}^{\mathrm{GDM}}) \rangle, \\
f(\boldsymbol{w}_t^{\mathrm{GDM}}) &\approx f(\boldsymbol{w}_{t-1}^{\mathrm{GDM}}) + \langle \boldsymbol{w}_t^{\mathrm{GDM}} - \boldsymbol{w}_{t-1}^{\mathrm{GDM}}, \nabla f(\boldsymbol{w}_{t-1}^{\mathrm{GDM}}) \rangle.
\end{aligned}
$$
Summing up the two equations gives
$$
\begin{aligned}
f(\boldsymbol{u}_t) &\approx f(\boldsymbol{w}_t^{\mathrm{GDM}}) + \langle \boldsymbol{u}_t - \boldsymbol{w}_t^{\mathrm{GDM}}, \nabla f(\boldsymbol{w}_{t-1}^{\mathrm{GDM}}) \rangle \\
&= f(\boldsymbol{w}_t^{\mathrm{GDM}}) + \frac{\mu}{1-\mu}\langle \boldsymbol{w}_t^{\mathrm{GDM}} - \boldsymbol{w}_{t-1}^{\mathrm{GDM}}, \nabla f(\boldsymbol{w}_{t-1}^{\mathrm{GDM}}) \rangle \\
&= f(\boldsymbol{w}_t^{\mathrm{GDM}}) - \frac{\mu}{1-\mu}\eta_{\mathrm{ef}} \left\langle (1-\mu)\sum_{s=1}^{t-1} \mu^{t-1-s}\nabla f(\boldsymbol{w}_s^{\mathrm{GDM}}) + \mu^{t-1}\nabla f(\boldsymbol{w}_1^{\mathrm{GDM}}), \nabla f(\boldsymbol{w}_{t-1}^{\mathrm{GDM}}) \right\rangle.
\end{aligned}
$$

Since
$$
\begin{aligned}
&- \eta_{\mathrm{ef}} \left\langle (1-\mu)\sum_{s=1}^{t-1} \mu^{t-1-s}\nabla f(\boldsymbol{w}_s^{\mathrm{GDM}}) + \mu^{t-1}\nabla f(\boldsymbol{w}_1^{\mathrm{GDM}}), \nabla f(\boldsymbol{w}_{t-1}^{\mathrm{GDM}}) \right\rangle \\
&= -(1-\mu)\eta_{\mathrm{ef}}\|\nabla f(\boldsymbol{w}_{t-1}^{\mathrm{GDM}})\|^2 + \mu \left\langle \boldsymbol{w}_{t-1}^{\mathrm{GDM}} - \boldsymbol{w}_{t-2}^{\mathrm{GDM}}, \nabla f(\boldsymbol{w}_{t-1}^{\mathrm{GDM}}) \right\rangle \\
&= -(1-\mu)\eta_{\mathrm{ef}}\|\nabla f(\boldsymbol{w}_{t-1}^{\mathrm{GDM}})\|^2 + \mu \left\langle \boldsymbol{w}_{t-1}^{\mathrm{GDM}} - \boldsymbol{w}_{t-2}^{\mathrm{GDM}}, \nabla f(\boldsymbol{w}_{t-2}^{\mathrm{GDM}}) \right\rangle + \mu \left\langle \boldsymbol{w}_{t-1}^{\mathrm{GDM}} - \boldsymbol{w}_{t-2}^{\mathrm{GDM}}, -\nabla f(\boldsymbol{w}_{t-1}^{\mathrm{GDM}}) + \nabla f(\boldsymbol{w}_{t-2}^{\mathrm{GDM}}) \right\rangle \\
&\approx -(1-\mu)\eta_{\mathrm{ef}}\|\nabla f(\boldsymbol{w}_{t-1}^{\mathrm{GDM}})\|^2 + \mu \left\langle \boldsymbol{w}_{t-1}^{\mathrm{GDM}} - \boldsymbol{w}_{t-2}^{\mathrm{GDM}}, \nabla f(\boldsymbol{w}_{t-2}^{\mathrm{GDM}}) \right\rangle + \mu H(\boldsymbol{w}_{t-2}^{\mathrm{GDM}}, \boldsymbol{w}_{t-1}^{\mathrm{GDM}} - \boldsymbol{w}_{t-2}^{\mathrm{GDM}}) \left\| \boldsymbol{w}_{t-1}^{\mathrm{GDM}} - \boldsymbol{w}_{t-2}^{\mathrm{GDM}} \right\|^2 \\
&\approx -(1-\mu)\eta_{\mathrm{ef}}\|\nabla f(\boldsymbol{w}_{t-1}^{\mathrm{GDM}})\|^2 + \mu \left\langle \boldsymbol{w}_{t-1}^{\mathrm{GDM}} - \boldsymbol{w}_{t-2}^{\mathrm{GDM}}, \nabla f(\boldsymbol{w}_{t-2}^{\mathrm{GDM}}) \right\rangle \\
&\approx \cdots \\
&\approx -\eta_{\mathrm{ef}}(1-\mu)\sum_{s=1}^{t-1} \mu^{t-1-s}\|\nabla f(\boldsymbol{w}_s^{\mathrm{GDM}})\|^2 - \eta_{\mathrm{ef}}\mu^{t-1}\|\nabla f(\boldsymbol{w}_1^{\mathrm{GDM}})\|^2 \\
&\approx -\eta_{\mathrm{ef}}\|\nabla f(\boldsymbol{w}_1^{\mathrm{GDM}})\|^2,
\end{aligned}
$$
and
$$
\begin{aligned}
f(\boldsymbol{u}_1) &\approx f(\boldsymbol{w}_1^{\mathrm{GDM}}) + \langle \nabla f(\boldsymbol{w}_1^{\mathrm{GDM}}), \boldsymbol{u}_1 - \boldsymbol{w}_1^{\mathrm{GDM}} \rangle + \frac{H(\boldsymbol{w}_1^{\mathrm{GDM}}, \boldsymbol{u}_1 - \boldsymbol{w}_1^{\mathrm{GDM}})}{2}\|\boldsymbol{u}_1 - \boldsymbol{w}_1^{\mathrm{GDM}}\|^2 \\
&= f(\boldsymbol{w}_1^{\mathrm{GDM}}) + \langle \nabla f(\boldsymbol{w}_1^{\mathrm{GDM}}), \boldsymbol{u}_1 - \boldsymbol{w}_1^{\mathrm{GDM}} \rangle + \frac{H(\boldsymbol{w}_1^{\mathrm{GDM}}, \boldsymbol{w}_2^{\mathrm{GDM}} - \boldsymbol{w}_1^{\mathrm{GDM}})}{2}\|\boldsymbol{u}_1 - \boldsymbol{w}_1^{\mathrm{GDM}}\|^2 \\
&\approx f(\boldsymbol{w}_1^{\mathrm{GDM}}) + \langle \nabla f(\boldsymbol{w}_1^{\mathrm{GDM}}), \boldsymbol{u}_1 - \boldsymbol{w}_1^{\mathrm{GDM}} \rangle \\
&= f(\boldsymbol{w}_1^{\mathrm{GD}}) - \eta_{\mathrm{ef}}\|\nabla f(\boldsymbol{w}_1^{\mathrm{GDM}})\|^2.
\end{aligned}
$$
As a conclusion, we have
$$f(\boldsymbol{w}_t^{\mathrm{GDM}}) \approx f(\boldsymbol{u}_t) + \eta_{\mathrm{ef}}\frac{\mu}{1-\mu}\|\nabla f(\boldsymbol{w}_1^{\mathrm{GDM}})\|^2 \approx f(\boldsymbol{w}_t^{\mathrm{GD}}) - f(\boldsymbol{w}_1^{\mathrm{GD}}) + f(\boldsymbol{u}_1) + \eta_{\mathrm{ef}}\|\nabla f(\boldsymbol{w}_1^{\mathrm{GDM}})\|^2 \approx f(\boldsymbol{w}_t^{\mathrm{GD}}).$$

The proof is completed.

