# OpenReview forum: "When and Why Momentum Accelerates SGD: An Empirical Study"
_ICLR.cc/2024/Conference — ICLR 2024 Conference Withdrawn Submission_

### Official Review · Reviewer_GteF · 2023-10-22

**Soundness:** 2 fair
**Presentation:** 1 poor
**Contribution:** 2 fair
**Rating:** 3
**Confidence:** 4

**Summary:**

This paper studies the performance gap between vanilla SGD and SGD with heavy-ball momentum (SGDM). Through experiments, the authors show that the number of iterations required to achieve a certain loss is similar for small learning rates, but for larger learning rates SGDM converges faster. This observation is somewhat consistent with different batch sizes. They also attempt to address why there is such gap, and attributes this gap to SGDM's lack of oscillation in training loss when using larger learning rates.

**Strengths:**

Since SGDM and other variants of momentum is often used in practice when training deep neural networks, the problem studied is of high relevance. The paper also takes into account many aspects of optimization recently gaining interest, such as EoS and sharpness.

**Weaknesses:**

Although the paper studies an interesting question, the submission requires significant improvement as the results are not very convincing. Below are a few major issues weakening this paper.

- I'm not convinced that using the effective learning rate stated in the paper is the right approach. There are two levels of approximations to get this effective learning rate: first, the gradients in the weighted sum is assumed to all be the same so that you can obtain the partial sum; then the final weight in the effective learning rate is taken to be the asymptotic value as $t\to\infty$. Is this really accurate and does it really make a more fair comparison for SGDM? Why not just do a grid search and compare the two methods under various step sizes? Using the effective step size this way also introduces a coupling between the two hyperparameters $\eta$ and $\mu$.
- (E2) in page 4: I don't agree with the sentence "If momentum's role is to cancel noise, then we would expect no effect of momentum in GD". For quadratics at least, without noise, GDM notably achieves acceleration both in theory and in practice. So shouldn't it be that "without noise, we *do* expect GDM to perform better than GD"? I think the authors are oversimplifying the "noise cancellation" effect of SGDM from previous works. This renders the motivation of this work somewhat weak.
- The paper claims to provide an answer to "why" SGDM performs better than SGD, but all the experiments are able to support is that there may be a correlation between SGD's oscillation and SGDM's lack of oscillation. The authors have made several conjectures and speculations, but the question itself is still inadequately addressed.
- There are some sentences that don't make sense to me, and I'm not sure what the authors are trying to claim. For instance:
	- top of page 6: "which implies some transition happens" - This is very vague. What transition? Transition from where to where?
	- "Increasing the batch size, the SGD and SGDM will diverge at a smaller effective learning rate". Perhaps you mean the gap between the two methods appears at a smaller effective learning rate? Also, I'm not sure if "acceleration of momentum interplays with learning rate and batch size" should be considered a new finding.
- I have some issues regarding the methodology used in the experiment presented in Figure 4 as well as the results. First of all, why is the deviation threshold measured in terms of "epochs to reach accuracy 0.99" but the two plots on the right are showing training loss? Shouldn't it be consistent? Moreover, the left plot displays "# of epochs to reach acc=0.99", so the lower the better. Yet the curve for GD is U-shaped and always above the curve for GDM. So as the effective learning rate increases, GD actually outperforms GDM when measured in accuracy? Does it not contradict results in the previous experiment? Furthermore, the training loss should be plotted in log-scale to better see what's happening in latter epochs.
- The third sentence in the statement of Proposition 1 does not make any sense. What is $\mathbf{A}$? I don't think this is ever defined. The proof of this proposition also directly assumes that $\mathbf{A}$ is diagonal, without saying what is actually is. Without explaining what this is, I do not understand what the Proposition is conveying, let alone evaluating the significance of the result and checking the correctness of the proof.


I also find the writing quality to be quite poor and overall the paper is not very organized. Some major issues here are:
- Table 1: This table is very difficult to parse and doesn't actually make comparing different works any easier. Presenting related works this way also runs the risk of inadequately summarizing the results of prior works. I would recommend writing a paragraph or so detailing the setup and results of each related work, putting more emphasis on the ones that matter the most to the current work.
- All figures should be in vectorized format and better presented. In Figure 3, it's extremely difficult to see what exactly is happening beyond effective learning rate $10^{-1}$ even when zoomed in, and one should always keep in mind that plots should be legible when printed on paper. Figure labels should also have their font size increased. It's very frustrating to parse results for an empirical paper when the plots are poorly presented.
- There are many grammatical errors and typos throughout the paper. I suggest the authors run a proofreading tool to improve the writing.

**Questions:**

- Figure 2 and 3: how come for $BS=1024$ and $\mu=0$, the curve has two fewer data points than with $\mu>0$? Is it because it diverges for the next effective learning rate? If so, it's common to still plot a data point by giving it the value of the maximum loss observed across the x-axis and indicate that at those point the loss has diverged.

**Minor issues**
- Abstract: please fix the quotation marks on "when"
- Page 2: "SGDM ~~experiences~~ approximately uses a larger effective..."

---

### Official Review · Reviewer_euxD · 2023-10-31

**Soundness:** 3 good
**Presentation:** 3 good
**Contribution:** 3 good
**Rating:** 6
**Confidence:** 3

**Summary:**

This paper tries to answer the questions of when and why SGD with momentum (SGDM) outperforms SGD through experiments.

For the question of when SGDM outperforms SGD, the key observation is that there is a threshold of effective learning rate which is affected by the batch size, and that SGDM outperforms SGDM only if the effective learning rate is beyond this threshold.

For the question of why SGDM outperforms SGD, the authors observe that the oscillation of GD happens exactly when it enters the EoF regime. They further conjecture that both momentum and stochastic noise help to defer the entrance of EoF. This conjecture is justified by both experiments and theoretical analysis.

**Strengths:**

1. The paper studies the acceleration of SGDM in an orderly manner, and the reasoning is very clear based on evidence.
2. The idea of the deferral of entering EoF as an effect of both momentum and stochastic noise is intriguing.
3. This explanation is a completely different one from the previous explanations, including counteracting high curvature and cancelling out noise.

**Weaknesses:**

1. The batch size used in the experiments in section 3 is either 1024 or 256, which is much larger than the commonly-used batch sizes (around 32). The authors could make the conclusions more convincing by studying these more practical cases.
2. It could be more convincing if the authors could use a proposition like Proposition 3 to justify that stochastic noise defers abrupt sharpening.
3. This phenomenon seems quite interesting because the scales of the spike in the directional Hessian of SGD with different batch sizes seem quite different (Figure 6B). It would be interesting to see more explanations on this.
4. Deferring the entrance of EoF as a combined effect of stochastic noise and momentum could be better illustrated by adding experiments with intermediate momentum parameter $\mu$ such that SGDM with intermediate batch size (i.e., batch size = 1000) does not enter EoF.

**Questions:**

1. According to the experiments, it seems that although SGDM outperforms SGD when the effective LR is larger than the threshold, their optimal performances are almost the same. Does it mean that fine-tuned SGDM can hardly outperform fine-tuned SGD?

---

### Official Review · Reviewer_PhKt · 2023-10-31

**Soundness:** 2 fair
**Presentation:** 2 fair
**Contribution:** 2 fair
**Rating:** 3
**Confidence:** 3

**Summary:**

This paper emperically studies when and why Momentum accelerates SGD.

**Strengths:**

- The authors conduct a fair comparison between SGD and SGDM by ensuring both algorithms operate with the same effective learning rate.
- This work introduces a novel perspective on the acceleration of momentum by examining the transitions between different phases of training dynamics.

**Weaknesses:**

- The experimental results can not sufficiently support some of the claims.
For example, the authors claim that ``For small $\eta_{\rm eff}$, SGDMs perform almost the same''. However, as shown in Figure 2, the authors only compare SGD and SGDM at the epoch 200. Notably, for smaller $\eta_{\rm eff}$, the sharpness-increasing phase must last longer, potentially causing acceleration to manifest not at epoch 200. However, the acceleration may appear at epoch 1000.


- Some of the explanations and results seem trivial.
  - Before entering EoS, the loss can effectively decrease. According to the linear stability theory in (Wu et al, 2018), the condition is $L_{GD}=2/\eta$. And for GDM, the stability condition $L_{GDM}>L_{GD}$, implyingthat the sharpness-increasing phase should last longer for GDM.
  - Additionally, Proposition 3.1 largely overlaps with the results in Cohen et al. (2021), and Proposition 3.3 largely overlaps with the results in Cohen et al. (2022).

- The paper lacks citations to earlier crucial works related to EoS, such as (Wu et al. (2018)) and (Jastrzebsk et al. (2020)).

[Wu et al]. How SGD selects the global minima in over-parameterized learning: A dynamical stability perspective. (2018)

[Jastrzebsk et al]. The break-even point on optimization trajectories of deep neural networks. (2020)

[Cohen et al]. Gradient descent on neural networks typically occurs at the edge of stability. (2021)

[Cohen et al].  Adaptive gradient methods at the edge of stability. (2022)

**Questions:**

- For SGDM, what are the underlying causes of "abrupt sharpening"? Notably, GD exhibits "progressive sharpening" as opposed to "abrupt sharpening" (Cohen et al, 2021).

- The loss can still descrese unstably at EoS (Ahn et al, 2022). Does the accelartion exists after enough long time?

[Cohen et al].  Gradient descent on neural networks typically occurs at the edge of stability. (2021)

[Ahn et al]. Understanding the unstable convergence of gradient descent. (2022)

---

### Official Review · Reviewer_Dpus · 2023-11-01

**Soundness:** 1 poor
**Presentation:** 3 good
**Contribution:** 2 fair
**Rating:** 3
**Confidence:** 4

**Summary:**

This paper tries to answer when and why momentum accelerates SGD based on empirical evaluations:

- For When: This paper observed that momentum accelerates SGD when the effective learning rate is larger than a certain threshold, and the threshold will decrease when increase the batch size.
- For Why: This paper showed that once the optimizer experiences abrupt sharpening (the directional Hessian experiences a sudden jump), the training process slows down and the momentum can significantly postpone the point of abrupt sharpening.

**Strengths:**

This paper is generally well written and identified some interesting empirical phenomenon about the effect of momentum on training neural networks.

**Weaknesses:**

I have some concerns on this work:

- The stochastic experiments (e.g., Figures 2,3, 6) showed training losses within each epoch, which is a biased estimation of the training loss over the full training dataset (since the losses were evaluated at different points), and is thus not so informative to the whole training progress and training stability. See Figure 2b and Section C in [3] for a discussion on training evaluation. The benefit of evaluating this loss is that it incurs no additional cost during training. However, this loss omits many convergence behaviors of an optimizer (see Figures 17 and 18 in the appendix of [3]).

- The stochastic experiments only run once, no deviation band is provided.

- Several empirical observations have already been made by previous works, which are not mentioned in the submission. E.g., [1] studied the effect of Nesterov's momentum with fixed effective learning rate (Section J), and by sweeping the momentum parameter, only limited improvement is observed over plain SGD. [2] studied the damping effect of momentum (Figure 1), which indicates that momentum made SGD more robust.

- The proofs to the propositions are not rigorous, many approximations are involved without formal justifications.


[1] Ma, Jerry, and Denis Yarats. "Quasi-hyperbolic momentum and Adam for deep learning." In ICLR, 2018.

[2] Lucas, James, et al. "Aggregated Momentum: Stability Through Passive Damping." In ICLR, 2018.

[3] Zhou, Kaiwen, et al. "Amortized Nesterov’s momentum: a robust momentum and its application to deep learning." In UAI, 2020.

**Questions:**

Please check the weaknesses section.